# Mental Health of Apprentices during the COVID-19 Pandemic in Austria and the Effect of Gender, Migration Background, and Work Situation

**DOI:** 10.3390/ijerph18178933

**Published:** 2021-08-25

**Authors:** Rachel Dale, Teresa O’Rourke, Elke Humer, Andrea Jesser, Paul L. Plener, Christoph Pieh

**Affiliations:** 1Department for Psychotherapy and Biopsychosocial Health, Danube University Krems, 3500 Krems, Austria; rachel.dale@donau-uni.ac.at (R.D.); teresa.orourke@donau-uni.ac.at (T.O.); elke.humer@donau-uni.ac.at (E.H.); andrea.jesser@donau-uni.ac.at (A.J.); 2Department of Child and Adolescence Psychiatry, Medical University of Vienna, 1090 Vienna, Austria; paul.plener@meduniwien.ac.at; 3Department of Child and Adolescent Psychiatry and Psychotherapy, University of Ulm, 89075 Ulm, Germany

**Keywords:** mental health, COVID-19, apprentices, gender, migration background, work situation

## Abstract

This study assessed the mental health of apprentices during the COVID-19 pandemic in Austria and the effect of gender, migration background, work situation, and work sector. An online survey via REDCap was performed with a sample of 1442 apprentices (female: 53.5%, male: 45.4%, diverse: 1.1%, migration background: 28.4%) from 29 March to 18 May 2021. Indicators of mental health were depression (PHQ-9), anxiety (GAD-7), well-being (WHO-5), disordered eating (EAT-8), and insomnia (ISI-7). There was a high prevalence of clinically relevant depression (cut-offs ≥11 for adolescents, ≥10 for adults: 48.3%), anxiety (cut-offs ≥11 for adolescents, ≥10 for adults: 35.4%), insomnia (cut-off ≥15: 27%), and disordered eating (cut-offs ≥2 for men, ≥3 for women: 50.6%). Linear models revealed that apprentices with female and diverse gender, migration background, and unemployed status showed the poorest scores on all mental health measures (all *p*-values < 0.05) except disordered eating. These findings emphasize the need for intersectional strategies to reduce and prevent adverse mental health consequences of the COVID-19 pandemic for apprentices.

## 1. Introduction

The COVID-19 pandemic and the associated measures to prevent an uncontrolled dissemination of the disease have impacted the lives of many people all over the world in all aspects of life. The Austrian government first introduced lockdown restrictions on 16 March 2020, with only few exceptions to enter public places, which ended on 30 April 2020, as COVID-19 cases decreased. As the cases started to rise again, further lockdown measures followed from 3 November 2020 to 8 February 2021. Such measures can negatively affect mental health [1,2], with consequences including increases in stress, depression, anxiety, and insomnia [3,4,5].

Mental health in Austria declined even after the first lockdown ended [6] and further declined in subsequent lockdowns [7]. Young people seem to be especially affected by these lockdown measures [7,8], and indeed, many international studies are beginning to emphasize the negative effects measures such as social distancing and home-schooling can have on children’s and adolescents’ mental health [9,10,11,12,13,14]. For example, a study conducted in May 2020 with 1794 Chinese adolescents reported an alarming prevalence as high as 36.7% for anxiety symptoms, 37.8% for insomnia, and 48.2% for depressive symptoms [15]. Reviews have summarized the adverse effects of the COVID-19 pandemic on adolescents’ mental health globally [16,17,18] and highlight the need for extensive prevention and management strategies for this age group. A recently published study on the mental health of Austrian school students aged 14 to 20 [19] revealed a high prevalence of mental disorders, such as depression, anxiety, insomnia, and eating disorders, one year after the outbreak of COVID-19, particularly in female and diverse participants.

Apprentices typically fall under the same age group as these school students, as apprenticeships usually start after finishing compulsory school around the age of 15, and indeed, the two groups are often compared [20], suggesting this sub-group of the population may have also faced similar challenges to high school students over the course of the pandemic. Apprenticeship refers to the Austrian system of dual vocational education and training in which a specific profession is learned by completing practical job training in a company as well as attending a vocational school to gain the necessary theoretical knowledge. An apprenticeship is completed after a final apprenticeship examination and is typically accomplished in two to four years depending on the profession. Work sectors in which apprenticeships are possible include, but are not limited to, trade, industry, commerce, banking and insurance, transport, information, and consulting or tourism. Therefore, although there are similarities with school students, being part of the work force rather than taking part in a purely educational environment means apprentices face some different experiences. For example, the transition from school to work is a developmental challenge that may present risk factors to mental health [21,22]. Indeed, a survey of Austrian apprentices from 2015 showed that apprentices assess their current state of health as significantly worse than high school students [23].

As such, assessing apprentice mental health separately from school students is warranted. However, to our knowledge, no studies have yet examined the mental health of apprentices in Austria during the COVID-19 pandemic. While school students faced challenges, such as home schooling [24] or class in shifts during the pandemic, many apprentices were affected by unemployment or furlough. Some work sectors, such as gastronomy or tourism, have been impacted by the lockdown measures and subsequent unemployment more than others [25].

A number of previous studies in young people have found poorer mental health in women than men during the pandemic [19,26], and therefore, we took gender into account during the analyses. Additionally, migration background has been found to be a risk factor for poor mental health in adolescents during the pandemic [14]. Austria has a high immigration rate—28% of 15–29 year-olds have a migration background [27]—and therefore, we investigated whether mental health differed according to migration background in this young apprentice sample. Finally, we predicted that those whose working situation changed the most and those working in the sectors most affected by the pandemic would show poorer mental health than those with more stable work conditions; specifically, those who lost their jobs or were put on furlough would show worse scores than those who continued working as before the pandemic, and those working in tourism/gastronomy or close-contact services (e.g. hairdressers) would have poorer mental health than other sectors. The aim of this study was to investigate how the mental health of apprentices in Austria has been affected by COVID-19 measures dependent on work situation and work sector as well as sociodemographic factors, such as gender and migration background, after one year of pandemic-related restrictions.

## 2. Materials and Methods

### 2.1. Study Design

This study was conducted from 29 March to 18 May 2021. On 1 April 2021, a regional lockdown was introduced in the eastern states of Austria (Vienna, Lower Austria, and Burgenland) due to rising COVID-19 cases and a critical number of patients in intensive care units in these states. A 24-h curfew was reinstated with the previous exceptions for leaving one’s home, such as covering basic needs, assistance and care of other people, work, and outdoor activities. For all other Austrian states, a curfew from 8 p.m. to 6 a.m. was in place, with the same exceptions. Further policies required wearing an FFP-2-mask in public indoor spaces and public transport and keeping a distance of at least two meters from other people. These tighter restrictions ended on 19 April in Burgenland and on 3 May in Vienna and Lower Austria.

Data were collected with an online survey via Research Electronic Data Capture (REDCap) [28], which is hosted at servers of the Danube University Krems, Austria. The survey was conducted online to enable rapid dissemination and completion and increase response rate, as it has been reported that respondents often prefer completion of questionnaires online [29]. Recruitment took place through various channels. The university posted information on its website as well as on social media. The Austrian Trade Union (Österreichischer Gewerkschaftsbund, ÖGB), of which around 40–45% of all Austrian apprentices are members, as well as the Austrian Chamber of Commerce (WKO) informed their members per e-mail and on social media. All participants had to agree to the data protection declaration (electronic informed consent), and all participants had to confirm that they were over the age of 14. The study was approved by the data protection officer and the Ethics Committee of the Danube University Krems (protocol code EK GZ 41/2018–2021), Austria, and conducted in accordance with the Declaration of Helsinki.

### 2.2. Measures

#### 2.2.1. PHQ-9

Depressive symptoms were measured with the German depression module of the Patient Health Questionnaire (PHQ-9) [30], which has been validated for adolescents [31]. It consists of nine self-report items on four-point scales ranging from 0 to 3, with a maximal total score of 27. To define clinically relevant depression, the 10-point cut-off score was used for adults (>18), and the 11-point cut-off score, as proposed by [32], was used for adolescents. Cronbach’s alpha was α = 0.88 in the current sample.

#### 2.2.2. GAD-7

The Generalized Anxiety Disorder (GAD-7) [33] scale was used to measure anxiety symptoms. It contains seven self-report items on four-point scales from 0 to 3 and a maximal total score of 21. Cut-off scores were set at 5 for mild, 10 for moderate, and 15 for severe anxiety symptoms. To define clinically relevant anxiety, the 10-point cut-off score was used for adults (>18), and the 11-point cut-off score was used for adolescents [34]. Cronbach’s alpha was α = 0.90 in the current sample.

#### 2.2.3. EAT-8

The Eating Attitudes Test (EAT-8) [35], a validated and reliable screening instrument, was used to measure symptoms of disordered eating. It contains eight items on a dichotomized response format with (1) “I agree somewhat” and (0) “I disagree somewhat”. The total scare can range from 0 to 8 and can be categorized into a low-risk group and a high-risk group. We used the 3-point cut-off score for female and the 2-point cut-off score for male adolescents, as recommended by Richter et al. [35]. Cronbach’s alpha was α = 0.83 in the current sample.

#### 2.2.4. ISI

The Insomnia Severity Index (ISI) [36] measures insomnia levels and sleep quality with seven items on four-point scales and has been validated for adolescents. The total score can be interpreted in following categories: no clinically significant insomnia (<7 points), sub-threshold insomnia (8–14 points), moderate clinical insomnia (15–21 points), and severe clinical insomnia (22–28 points). To define clinically relevant insomnia, we used the 15-point cut-off score. Cronbach’s alpha was α = 0.86 in the current sample.

#### 2.2.5. WHO-5

The WHO-5 questionnaire [37] measures well-being with five self-report items on six-point Likert scales. It has a score range from 0 (no well-being) to 25 (optimal well-being) and good psychometric properties [38]. Cronbachs alpha was α = 0.87 in the current sample.

#### 2.2.6. Other Variables

Gender was coded as female, male, or diverse. Migration background was coded as yes/no and was considered yes if the participant and/or both parents were born abroad. Work situation consisted of five categories: as before (situation unchanged from before the pandemic), furlough (the Austrian ‘Kurzarbeit’ scheme, whereby working hours are reduced, and the government supports salaries), home office (home office has increased during the pandemic), lost job (since the start of the pandemic), and mixed (a mixture of the aforementioned categories). Work sector was broken down into nine categories: trade and craft, industry, retail, banking/insurance, transport/traffic, gastronomy/tourism, information/consulting, close-contact services (e.g., hairdressers, massage, which sometimes had different COVID-19 regulations to other industries), and other (examples here included administration and gardener).

### 2.3. Statistical Analyses

There were significant differences in working sector and working situation according to gender (χ^2^(16) = 174.16, *p* < 0.0001; χ^2^(8) = 43.12, *p* < 0.0001, respectively) and migration background (χ^2^(8) = 65.64, *p* < 0.0001; χ^2^(4) = 44.54, *p* < 0.0001, respectively). Furthermore, working situation differed according to the area of work (χ^2^(32) = 357.06, *p* < 0.001). Therefore, to control for the influence of these variables on each other, linear models were run with the mental health measures (PHQ-9, GAD-7, ISI, EAT-8, and WHO-5 total scores) as the dependent variable and working situation, working sector, gender, and migration background as factors. For investigating the proportion of respondents above the cut-off scores for clinically relevant depression, anxiety, insomnia, and disordered eating, binomial general linear models were run. All analyses were conducted in R version 4.0.3 [39]. Post-hoc tests were conducted using the estimated means from the models in the emmeans package [40].

## 3. Results

### 3.1. Sample

From a total of 2945 individuals who visited the informed consent page of the survey, 2535 gave informed consent and confirmed to be at least 14 years old, resulting in a participation rate of 86.1%. A total of 1659 apprentices completed the survey, resulting in a completion rate of 65.4%. From the 1659 participants who completed the survey, 1442 (86.9%) were included in the final analyses. The remaining 217 (13.1%) failed an attention check item and were therefore excluded. The final sample consisted of 1442 Austrian apprentices (female: 53.5%, male: 45.4%, diverse: 1.1%, migration background: 28.4%) aged 15 to 43 years (M = 18.19, SD = 2.30). Work situation and work sector characteristics of the study sample are presented in Table 1.

Descriptive statistics for each mental health measure for the total sample and according to gender, migration background, and work situation can be seen in Table 2. In the total sample, 48.3% were over the cut-off for moderate depression, 35.4% for moderate anxiety, 27% reported clinically relevant insomnia, and 50.6% were over the cut-off for disordered eating behavior. Table 3 shows the percentage of participants over the cut-offs for depression, anxiety, disordered eating, and insomnia according to gender, migration background, and work situation.

### 3.2. Depression

PHQ-9 scores were significantly affected by gender (F(2) = 64.87, *p* < 0.0001), working situation (F(4) = 5.24, *p* < 0.001), and migration background (F(1) = 4.09, *p* < 0.05) but not by working sector (F(8) = 1.73, *p* = 0.09). Women showed poorer depression scores than men (*p* < 0.0001), and diverse participants showed poorer scores than both women (*p* < 0.01) and men (*p* < 0.0001). Those who lost their jobs showed higher depression scores than those who continued working as before (*p* = 0.0001), those on furlough (*p* < 0.05), those with increased home office (*p* < 0.001), and those with a mixed situation (*p* < 0.05). Those with a migration background had higher scores than those without (*p* < 0.05).

Being over the cut-off for clinically relevant depression was significantly affected by gender (χ^2^(2) = 90.67, *p* < 0.0001) and work situation (χ2(4) = 10.70, *p* = 0.03) but not migration background (χ^2^(1) = 1.13, *p* = 0.29) or work sector (χ^2^(8) = 8.31, *p* = 0.4). Women and diverse participants were more likely to be over the cut-off than men (*p* < 0.0001 and *p* < 0.01 respectively). Those who lost their jobs were more likely to be over the cut-off than those working as before and those with increased home office (both *p* < 0.05).

### 3.3. Anxiety

GAD-7 scores significantly differed by gender (F(2) = 52.61, *p* < 0.0001) and work situation (F(4) = 3.62, *p* < 0.01), and was marginally affected by migration background (F(1) = 3.69, *p* = 0.06) but was not affected by working sector (F(8) = 1.25, *p* = 0.27). Women showed higher anxiety scores than men (*p* < 0.0001), and diverse participants showed poorer scores than both women (*p* < 0.05) and men (*p* < 0.0001). Those who lost their jobs showed higher anxiety scores than those who continued working as before (*p* = 0.03) and those with increased home office (*p* < 0.01).

Being over the cut-off for clinically relevant anxiety significantly differed according to gender (χ^2^(2) = 68.99, *p* < 0.0001) and work situation (χ^2^(4) = 9.43, *p* = 0.05) but not migration background (χ^2^(1) = 1.49, *p* = 0.22) or work sector (χ^2^(8) = 6.16, *p* = 0.63). Women and diverse participants were more likely to be over the cut-off than men (*p* < 0.0001 and *p* < 0.001 respectively). Those who lost their jobs were more likely to be over the cut-off than those with increased home office (*p* < 0.05).

### 3.4. Well-Being

WHO-5 scores differed significantly according to gender (F(2) = 56.36, *p* < 0.0001), working situation (F(4) = 4.43, *p* = 0.001), and migration background (F(1) = 4.93, *p* < 0.05) but not work sector (F(8) = 1.81, *p* = 0.07). Women showed lower well-being than men (*p* < 0.0001), and diverse participants showed poorer scores than both women (*p* = 0.02) and men (*p* < 0.0001). Those who lost their jobs showed poorer well-being than those who continued working as before (*p* = 0.01) and those with increased home office (*p* < 0.01).

### 3.5. Disordered Eating

EAT-8 scores were only affected by gender (F(2) = 25.74, *p* < 0.0001) and not by working situation (F(4) = 1.78, *p* = 0.13), migration background (F(1) = 0.63, *p* = 0.43), or work sector (F(8) = 0.33, *p* = 0.95). Women had higher disordered eating scores than men (*p* < 0.0001).

There were no effects of gender (χ^2^(2) = 1.83, *p* = 0.4), work situation (χ^2^(4) = 3.17, *p* = 0.53), migration background (χ^2^(1) = 1.51, *p* = 0.22), or work sector (χ^2^(8) = 5.84, *p* = 0.67) on the likelihood of being over the cut-off for disordered eating.

### 3.6. Sleep

ISI scores were significantly affected by gender (F(2) = 37.37, *p* < 0.0001), working situation (F(4) = 3.94, *p* < 0.01), and migration background (F(1) = 17.18, *p* < 0.0001) but not by working sector (F(8) = 1.58, *p* = 0.13). Women and diverse participants had higher insomnia scores than men (*p* < 0.0001 and *p* < 0.01, respectively). Those who lost their jobs had poorer sleep than those working as before (*p* = 0.01) and those with increased home office (*p* < 0.01). Those with a migration background had higher insomnia scores compared to those without (*p* < 0.0001).

Being over the cut-off for clinically relevant insomnia significantly differed according to gender (χ^2^(2) = 36.4, *p* < 0.0001), work situation (χ^2^(4) = 12.22, *p* < 0.05), and migration background (χ^2^(1) = 5.44, *p* < 0.05) but not work sector (χ^2^(8) = 12.45, *p* = 0.13). Women and diverse participants were more likely to be over the cut-off than men (*p* < 0.0001 and *p* < 0.01, respectively). Those who lost their jobs were more likely to be over the cut-off than those working as before and those with increased home office (*p* < 0.05 and *p* = 0.01 respectively).

## 4. Discussion

Overall, this sample of Austrian apprentices have experienced relatively poor mental health during the COVID-19 pandemic. In the total sample, 48% were over the cut-off for clinically relevant depressive symptoms, 35% for anxiety, 50% for disordered eating, and 27% for insomnia. These results are comparable to findings on other young populations in Austria. For example, in a study conducted in February 2021, 55% of school aged adolescents were over the depression cut-off, 47% for anxiety, 23% for insomnia, and 59% for disordered eating [19]. Similar scores were reported in 18–24-year-old adolescents surveyed over the Christmas period in Austria (depression: 50%, anxiety: 35%, insomnia: 25% [7]).

An important aspect for apprentices during this pandemic is how much their working situation had to change due to restrictions, as this can have implications for learning, networking, and future career opportunities. As anticipated, work situation had an effect on mental health, with unemployed participants having the poorest depression, anxiety, insomnia, and well-being scores. This finding is in line with current research on COVID-19-related unemployment and mental health. Unemployment was identified as a risk factor for developing mental health problems during the first lockdown in Austria [8]. Moreover, an early study conducted in Israel reported that unemployment during the COVID-19 pandemic was associated with increased psychological distress in young people [41]. Similarly to our results, a study conducted with a representative national sample from South Africa revealed that individuals with continued paid employment during lockdown had significantly lower depression scores than individuals that lost their employment as a consequence of lockdown measures [42]. Furthermore, a recently published study with a representative national sample from the United States [43] determined a higher probability of developing depression and anxiety when involuntarily unemployed during the COVID-19 pandemic than when staying employed or having voluntarily terminated work. Additionally, the same study found job uncertainty to be a contributing factor for the subsequent development of mental disorders. This is a major factor to consider for apprentices, as their work situations are generally less secure than those of most adults with a finished education, and established careers and finding a new job after unemployment is more difficult without final job qualifications.

Youth unemployment numbers averaged at 43,453 adolescents and young adults aged 15 to 24 who registered as unemployed with the Austrian national unemployment office AMS (Arbeitsmarktservice) in the year 2020, which constitutes an increase of 43.6% in comparison to the year 2019 [44]. Despite an 8.2% decrease of started apprenticeships from 2019 to 2020, a total of 8159 adolescents and young adults were on the search for an apprenticeship in Austria in the year 2020, which is an increase of 19.5% in comparison to the year before [44]. As unemployment, especially at a young age, can have detrimental consequences for mental and physical health [45,46] as well as for future income [47], these numbers are alarming. As of May 2021, the number of adolescents and young adults registered as unemployed with the AMS (excluding those in job training at the AMS) has decreased by 52.4% in comparison to May 2020 [48]. Nevertheless, COVID-19-related youth unemployment constitutes a challenge with far-reaching consequences for many sectors and requires intersectional solutions. An exemplary EU-wide approach to counteract these adverse developments has been proposed [49].

There are large differences in how work sectors in Austria have been affected by the pandemic and related measures. For example, gastronomy and tourism was the most affected work sector, with only 7.4% of all acquisitive individuals being employed, 43.4% on furlough, and 35% unemployed after the first lockdown in March 2020 [50]. In comparison, the finance and insurance sector was much less affected by restrictions, with 88.7% of all acquisitive individuals being employed, 7.5% on furlough, and approximately only 1% unemployed after March 2020 [50]. However, work sector had no stand-alone effect on mental health when controlling for the effect of work situation. This suggests that differences in mental health in these work sectors are likely due to the different work situations resulting from varying restrictions.

As expected, female and diverse participants showed poorer mental health than males on all measures, with the exception of likelihood of being over the cut-off for disordered eating (although women had a higher mean score than men). The finding that men show better mental health than other genders is common both within [26] and outside of the COVID-19 pandemic [51], including in younger populations [19], and demonstrates that the gender gap in mental health is already clear in adolescents and young adults. However, this effect seems to have increased in times of the pandemic (e.g., [7]).

In addition, we found an effect of having a migration background on a number of mental health measures; in those with a migration background, mean depression, anxiety, and insomnia scores were higher; well-being scores were lower; and the likelihood of being over the cut-off for clinically relevant insomnia was higher than in those without a migration background. Aside from the pandemic, a migration background can be a risk factor for poor mental health in and of itself, especially in young people [52,53,54]. This has been explained by the impact of the migration experience [55], increased economic hardship in migrant families, the experience of social exclusion and discrimination [53,56], and difficulties accessing health care services [53,57]. Some of these circumstances are likely to have been exacerbated by the pandemic. Poor living conditions due to low income levels may have made it harder to endure curfews or work from home. Existing financial difficulties may have increased fears of job loss and wage cuts. Many low-threshold psychosocial services could not be continued during the lockdowns (i.e., street work, outreach services, etc.), which may have resulted in even fewer support services being available in some places. Austria has a relatively high proportion of residents with a migration background (23.7% of the population, first or second generation), and the average age of foreign nationals is younger than that of domestic citizens [58]. Therefore, apprenticeships are likely to be an important avenue into the work force for many foreign nationals, and the current results indicate that the mental health of this group requires particular attention.

There are some limitations to the study that warrant mention. The survey was cross-sectional, preventing any causal conclusions regarding the COVID-19 pandemic and apprentice mental health. Furthermore, the sample sizes of some sub-groups, namely diverse participants and those who lost their jobs, are rather small, and therefore, caution must be taken when drawing conclusions from these groups. Although apprentices typically refer to younger people in their late teens/early 20 s, it should be noted our sample included a few older participants (>26 years, *N* = 14). However, the mean age was 18, with a standard deviation of 2.3, and therefore, it is unlikely these respondents had an impact on the results. Carrying out the survey online is another shortcoming that might have introduced sources of bias, such as not capturing apprentices who lack access to the internet or those who cannot proficiently use technology. Lastly, the large-scale online nature of the study, although allowing for a large sample collected without direct social contact, only included self-reported scores of mental health measures. Structured clinical interviews would provide a more objective measure of mental health, and related to this, we could not control for pre-existing mental health conditions.

## 5. Conclusions

In sum, apprentice mental health one year into the COVID-19 pandemic is comparable to that of school aged-adolescents and young adults in Austria. Those who lost their apprenticeship seem to be particularly burdened, as well as female apprentices and those with a migration background, which highlights the need for intersectional approaches to reduce the adverse consequences of the COVID-19 pandemic on the mental health of Austrian apprentices. It will be important to develop opportunities for apprentices during this crucial period of their lives to make up for the disruption to their career entry over the last year.

## Figures and Tables

**Table 1 ijerph-18-08933-t001:** Work situation and work sector characteristics of the study sample (*N* = 1442).

Variable	*N*	%
Total	1442	100
Work situation		
As before	872	60.5
Home office	378	26.2
Furlough	74	5.1
Lost job	26	1.8
Mixed *	92	6.4
Work sector		
Trade	334	23.2
Industry	279	19.3
Commerce	223	15.5
Bank/insurance	77	5.3
Transport/traffic	51	3.5
Gastronomy/tourism	66	4.6
Information/consulting	75	5.2
Close-contact services †	44	3.1
Other	293	20.3

* a mixture of some or all of the other work situation categories. † services involving close body contact, such as hairdressers.

**Table 2 ijerph-18-08933-t002:** Means and standard deviations of each mental health measure for the total sample and according to gender, migration background, and work situation.

	Depression—PHQ-9	Anxiety— GAD-7	Well-Being— WHO-5	Disordered Eating—EAT-8	Sleep— ISI
TOTAL	10.59 (6.57)	8.28 (5.49)	47.55 (22.80)	2.77 (2.51)	10.05 (6.52)
*Gender*					
Female	12.13 (6.55)	9.48 (5.47)	42.44 (22.21)	3.18 (2.58)	11.31 (6.49)
Male	8.61 (6.02)	6.75 (5.10)	54.06 (21.87)	2.26 (2.32)	8.48 (6.21)
Diverse	17.31 (5.52)	12.69 (5.19)	27.50 (13.22)	3.56 (2.28)	13.81 (5.91)
*Migration background*					
Yes	11.11 (6.48)	8.69 (5.36)	45.56 (23.69)	2.85 (2.46)	11.12 (6.35)
No	10.37 (6.60)	8.11 (5.54)	48.37 (22.39)	2.73 (2.52)	9.61 (6.54)
*Work situation*					
As before	10.07 (6.34)	7.99 (5.43)	49.24 (22.09)	2.69 (2.51)	9.58 (6.42)
Home office	10.75 (6.80)	8.23 (5.48)	47.04 (23.48)	2.82 (2.52)	10.26 (6.47)
Furlough	12.08 (6.57)	9.32 (5.39)	43.78 (24.09)	3.45 (2.71)	11.70 (6.73)
Lost job	16.08 (6.53)	11.85 (5.03)	32.77 (23.43)	3.23 (2.58)	14.54 (6.60)
Mixed *	12.05 (6.78)	9.37 (5.84)	40.83 (22.71)	2.62 (2.16)	11.13 (6.76)

PHQ-9, Patient Health Questionnaire 9 scale; GAD-7, Generalized Anxiety Disorder 7 scale; WHO-5, Well-being questionnaire of the World Health Organization; EAT-8, Eating Attitudes Test 8; ISI, Insomnia Severity Index. * a mixture of some or all of the other work situation categories.

**Table 3 ijerph-18-08933-t003:** Percentage of respondents over the cut-off for clinically relevant mental health disorders according to gender, migration background, and work situation.

	Depression—PHQ-9 Cut-Off 10/11	Anxiety— GAD-7 Cut-Off 10/11	Disordered Eating—EAT-8 Cut-Off 2/3	Sleep— ISI7 Cut-Off 15
TOTAL	48.3%	35.4%	50.6%	27%
*Gender*				
Female	59.3%	44.6%	52.4%	33.9%
Male	34.2%	23.5%	48.4%	18.2%
Diverse	93.8%	75%	56.3%	56.3%
*Migration background*				
Yes	50.7%	38.1%	53.1%	31.9%
No	47.3%	34.2%	49.6%	25%
*Work situation*				
As before	44.8%	34.2%	49.7%	24.7%
Home office	50.5%	34.9%	51.3%	27.2%
Furlough	56.8%	35.1%	59.5%	33.8%
Lost job	80.8%	65.4%	53.8%	57.7%
Mixed *	55.4%	40.2%	48.9%	33.7%

PHQ-9, Patient Health Questionnaire 9 scale; GAD-7, Generalized Anxiety Disorder 7 scale; WHO-5, Well-being questionnaire of the World Health Organization; EAT-8, Eating Attitudes Test 8; ISI, Insomnia Severity Index. * a mixture of some or all of the other work situation categories.

## Data Availability

Data are available upon request.

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
