# Peer review of "Mental Health of Apprentices during the COVID-19 Pandemic in Austria and the Effect of Gender, Migration Background, and Work Situation"

_ijerph, 2021, doi:10.3390/ijerph18178933_

Round 1

Reviewer 1 Report

The goal of this study was to investigate how the mental health of apprentices in Austria has been affected by COVID-19 measures dependent on work situation and work sector, as well as sociodemographic factors such as gender and migration background, after one year of pandemic-related restrictions. This is a timely and generally well-written paper. There are a number of questions that arise in the reading of this paper that are reported below.

1-Line 65 to 68. “The aim of this study….” should be moved to the end of the introductory chapter.

2- Line 103: How did you define your sample size and patient recruitment and its stratification (ages, gender, professional category, etc ...?

3-Line 201 to 260:  with a table the results would be easier to read.

Reviewer 2 Report

Congratulations on your research on the mental health of apprentices in Austria in the covid pandemic.
Here are some comments on the different sections of the manuscript:
Introduction: You introduce very well the research topic and the need to study apprentices separately from other students of their age. However, I think it could benefit from some more articles on mental health in young people in other countries or globally during the pandemic.
Material and methods: The weakness of the study is the online survey, due to the lack of representativeness of the sample inherent in this methodology. It is necessary to include more details (how the information was disseminated so that trainees could access the survey, how all stakeholders were reached, what percentage answered all the questionnaires in relation to the total of those who started in order to know the losses, etc...) and also to include theoretical justification as to why this type of online questionnaire was the only way to carry out the study at that particular moment in time.
The results about the sample are included in this section, I have doubts whether they should be here or at the beginning of the results section.
Results: Very well presented. Perhaps what I like the least is the Figure. I don't think it is necessary having the tables and the wording of the results.
Discussion: Well written, answers the research questions. But I think it could benefit from some more articles to compare the results in Austria with those in other countries, for example in relation to job loss, even in other older age groups. It seems that the socio-economic consequences of the pandemic (including job loss) strongly affect mental health, in all age groups. 
Conclusions: They answer the research question in line with the results presented previously. 

Round 2

Reviewer 2 Report

The changes have improved the quality of the manuscript. Further changes are not required. Congratulations.